# Clinical Effects of the Immunization Protocol Using *Loxosceles* Venom in Naïve Horses

**DOI:** 10.3390/toxins14050338

**Published:** 2022-05-13

**Authors:** Ana Luísa Soares de Miranda, Bruno Cesar Antunes, João Carlos Minozzo, Sabrina de Almeida Lima, Ana Flávia Machado Botelho, Marco Túlio Gomes Campos, Carlos Delfin Chávez-Olórtegui, Benito Soto-Blanco

**Affiliations:** 1Department of Veterinary Clinics and Surgery, Veterinary College, Federal University of Minas Gerais, Belo Horizonte 30123-970, MG, Brazil; analuisa.miranda@hotmail.com (A.L.S.d.M.); camposmtg@gmail.com (M.T.G.C.); 2Department of Health of the State of Paraná, Production and Research Center of Immunobiologicals, Piraquara 80230-140, PR, Brazil; bruno.antunes@sesa.pr.gov.br (B.C.A.); joao.minozzo@sesa.pr.gov.br (J.C.M.); 3Department of Biochemistry and Immunology, Institute of Biological Sciences, Federal University of Minas Gerais, Belo Horizonte 31270-901, MG, Brazil; sabrina.lima79@yahoo.com.br (S.d.A.L.); olortegi@icb.ufmg.br (C.D.C.-O.); 4Department of Veterinary Medicine, Veterinary College, Federal University of Goiás, Campus Samambaia, Goiânia 74690-900, GO, Brazil; anafmb@ufg.br

**Keywords:** antisera production, brown spider, dermonecrosis, loxoscelism, safety evaluation

## Abstract

Bites of brown spiders (*Loxosceles* spp.) are responsible for dermonecrotic lesions and potentially systemic envenoming that can lead to death. The only effective therapy is the use of the antivenom, usually produced in horses. However, little is known about the consequences of the systematic use of the *Loxosceles* venom and adjuvants and of the bleedings on antivenom-producing horses. Thus, the aim of this study was to evaluate the clinical changes in horses in their first immunization protocol for *Loxosceles* antivenom production. Eleven healthy horses, never immunized, were evaluated in three different periods: T0 (before immunization); T1 (after their first venom immunization); and T2 (after their first bleeding). Horses were clinically evaluated, sampled for blood, and underwent electrocardiographic (ECG) recordings. Several suppurated subcutaneous abscesses occurred due to the use of Freund’s adjuvants and thrombophlebitis due to systematic venipunctures for the bleeding procedures. ECG showed arrhythmias in few horses in T2, such as an increase in T and R waves. In summary, the immunization protocol impacted on horses’ health, especially after bleeding for antivenom procurement.

## 1. Introduction

Spiders of the genus *Loxosceles*, known as brown spiders, are found worldwide. Their bites are responsible for severe dermonecrotic lesions around the bite site in humans and animals, but systemic toxicity may also occur, including hemolysis, hemostatic disturbances, and acute renal failure [1,2,3,4]. *Loxosceles* venom is composed of several toxins, including sphingomyelinases D (or phospholipases D), hyaluronidases, astacin-like metalloproteases, translationally controlled tumor protein (TCTP), and *Loxosceles* allergen-like toxin (LALLT). The main toxins are the sphingomyelinases D, that cleave tissue phospholipids, enhancing the inflammation and the local tissue injury [5,6]. Hyaluronidases boost the spread of venom toxins [7], astacin-like metalloproteases hydrolyze the extracellular matrix [8], and TCTP and LALLT promote the release of histamine [9,10]. Several other toxins are also present in the *Loxosceles* venom, but their toxic action is not fully understood [6].

Antivenom remains the only effective treatment against this envenomation [11,12]. Horses are the most used species for antivenom production around the world. One of the obstacles for antivenom production is the welfare of horses submitted to immunization protocols [12,13]. In fact, the crude venoms used in the immunization protocols do not undergo any detoxification process, thus assuring that no structural or epitopic modification occurs that can alter its immunogenic capacity [14]. However, little is known about the consequences of the systematic use of the *Loxosceles* venom and adjuvants and of the blood collections on antivenom-producing horses. Thus, the aim of this study was to evaluate the clinical changes in horses in their first immunization protocol for *Loxosceles* antivenom production.

## 2. Results

### 2.1. Clinical Examination

Animals were alert during all three evaluation periods, presenting no alteration in general behavior. All horses presented mucous membranes with standard coloration (healthy, pink, and moist) during the first two evaluation periods, but pale mucous membranes were observed in six horses after first bleeding (T2). All animals presented abscess in the infiltration sites of neck and gluteal regions in T1, with an increase in size and number and suppuration in T2 (Figure 1), as the inoculation was being made in more sites. Thrombophlebitis of the external jugular vein was observed in five horses in T2.

The body weights, pulse and respiratory rates, and rectal temperature are presented in Table 1. No statistical difference was observed for all these parameters in the three evaluation periods.

### 2.2. Blood Parameters

CBC results are portrayed in Table 2. RBC, hemoglobin, and PCV showed progressive reduction (*p* < 0.05) after first venom immunization (T1) and first bleeding (T2). WBCs were higher (*p* < 0.05) after first venom immunization (T1) and first bleeding (T2) than before immunization (T0); the mean counts in both times were above the reference values. Lymphocyte counts were reduced (*p* < 0.05) just after first venom immunization (T1), whereas platelet counts were increased (*p* < 0.05). PDW, MPV, and P-LCR showed significant differences (*p* < 0.05) in all three evaluation periods.

The results of the blood biochemical panel are presented in Table 3. Creatine levels and GGT activities were reduced (*p* < 0.05) after first bleeding (T2). AST activities and albumin and cholesterol levels were lower (*p* < 0.05) after first venom immunization (T1) and first bleeding (T2) than before immunization (T0). Glucose levels were reduced (*p* < 0.05) just after first venom immunization (T1). Triglycerides were increased (*p* < 0.05) after first bleeding (T2).

### 2.3. Cardiac Examinations

The cardiac auscultation after first commercial bleeding (T2) showed splitting of the first (S1) or second (S2) heart sound in eight horses. No heart murmur was found at T0 and T1.

ECG results are presented in Table 4. Eight horses presented a bifid P wave (Figure 2A) at all evaluations. Horses presented normal sinus as the predominant rhythm (Figure 2A), as well as sinus arrhythmia, and sinus tachycardia (Figure 2B). Arrhythmias were found in two horses only in T2, consisting of an advanced second-degree atrioventricular (AV) block (Figure 2C) in which normal QRS complexes are interspersed by periods where P waves are not followed by QRS complexes, although maintaining PP intervals.

HR in T2 showed higher (*p* < 0.05) values compared to T1 but means of all evaluated periods were above the reference values for the species. P (mV) showed reduction (*p* < 0.05) just after first venom immunization (T1), whereas QT were increased (*p* < 0.05). T and R waves were increased (*p* < 0.05) after first venom immunization (T1) and first bleeding (T2). Intervals of PR and QRS showed no statistical difference between evaluation periods and were kept within the reference values for the species in all periods.

## 3. Discussion

The major finding during the visual inspection was the presence of subcutaneous abscesses. In the current immunization protocol, Freund’s complete adjuvant is used on the first day of the hyperimmunization cycle. This adjuvant is responsible for granulomas, abscesses, and ulcerative necrosis at the injection site [16], such as that observed in the present study. However, it is also feasible to assume that the *Loxosceles* venom would cause the subcutaneous abscesses.

Immunization protocols impose a catabolic effect on horses’ metabolism since high amounts of energy and protein mobilizations are made to produce globulins [17]. In the present study, horses demanded about 221 days to achieve the total of specific antibodies desired for antivenom production. The absence of weight loss confirms that nutritional management was correctly conducted, which is a major factor that addresses animal welfare regulations.

One relevant clinical change observed in the horses of this study was thrombophlebitis of the external jugular vein, a common clinical complication in equine internal medicine. Thrombophlebitis is defined by a vein thrombosis with concurrent inflammation [18,19,20,21]. In the present study, thrombophlebitis probably developed due to the use of this vein for the repeated bleeding required for antivenom production.

Leukocytosis was observed in the horses of this study after first venom immunization (T1) and first bleeding (T2), probably indicating an inflammatory response due to the use of adjuvants. Adjuvants are intended to promote an inflammatory response boosting the immune response and consequently enhancing antibody titers [22,23]. The use of Freund’s complete adjuvant in the present study was able to induce ulcers and granulomas, an effect previously reported [16], which can also explain the neutrophilia observed. Similar results were observed in horses injected with Freund’s complete adjuvant [24] and a horse used for snake antivenom production using Freund’s adjuvant [14] or Montanide [17].

The bleeding protocol used in the present study caused anemia in horses, evidenced by the progressive reduction and RBC, hemoglobin, and PCV and the observed pale mucous membranes in six of eleven horses after first bleeding (T2), which was similar to findings described earlier [14,25]. Two horses kept PCV under 15% during T2, which shows a clinical concern since, in most of these cases, a blood transfusion is required [26].

Platelet counts were inferior to reference values after first bleeding (T2), which may be linked to the bleeding for obtaining the antiserum. PDW and MPV showed significant differences in all three evaluation periods. A direct relation between PDW and MPV is described in healthy humans. This relation is kept in patients with thrombocytopenia derived from peripheric destruction [27], such as the one observed in horses of the present study due to the partaking in bleeding procedures. If thrombocytopenia was derived from a production deficit, an increase in PDW and a decrease in MPV would be observed, since PDW depends on the platelet production by megakaryocyte fragmentation [27].

In horses of this study, albumin levels presented significant reduction during the evaluation periods, probably due to repeated bleeding. Globulins sustained a pattern of increase during all evaluation periods, which was expected since the inoculated antigens and adjuvants employed in the immunization protocols are responsible for inducing the synthesis of immunoglobulins [28].

A study using mice, isolated perfused heart preparations, and ventricular cardiac myocytes showed *L. intermedia* venom might exert a direct cardiotoxic effect [29]. No significant cardiac alterations were found in T0 and T1; however, in T2, eight horses presented heart murmurs during cardiac auscultation, characterized by splitting of the first (S1) or second (S2) heart sound. S1 occurs at the beginning of ventricular systole and represents the closure of atrioventricular valves. S2 splitting is an asynchronous closure of semilunar valves and is considered physiological in horses [30]. Variation in the intensity and quality of S1 is uncommon in horses [31], but bleeding of such magnitude might justify this finding. The S1 split can come from a prolonged diastolic period and arrhythmias (such as the ones diagnosed in the present study) [32].

ECG recordings were performed because arrhythmias can be diagnosed by clinical examination but might require complementary exams to reach a reliable diagnosis [15,33,34]. T and R waves were statistically higher after venom immunization and bleeding than before immunization. T wave represents ventricle repolarization, and its alterations can often be related to electrolyte imbalance (such as hyperkalemia) as well as coronary vasospasm [35]. An increase in alpha-adrenergic receptor activity in epicardial coronary arteries or excessive catecholamine release that may activate these receptors can lead to coronary vasospasm [36]. An increase in R waves may be caused by myocardial damage (such as ischemia) or tachycardia since both alter electric heart conduction. Another finding that corroborates with myocardial ischemia is a biphasic T wave [37], but several biphasic T waves were also recorded before immunization, prior to injection of *Loxosceles* venom and bleedings. These changes may be promoted by any cardiotoxic effect of the loxoscelic venom, as reported earlier [23,29], or by any effect of the adjuvants.

## 4. Conclusions

The *Loxosceles* venom immunization and bleeding protocols impacted horses’ health, especially after bleeding for antivenom procurement. Thrombophlebitis and subcutaneous abscesses were the main clinical findings, but are inherent to antivenom procurement management, due to the systematic venipunctures and use of adjuvants at the venom inoculation sites, respectively. Other effects were anemia and slight cardiac and metabolic changes. New studies must be performed aiming to develop an immunization protocol with higher safety for the animals used for *Loxosceles* antivenom production.

## 5. Materials and Methods

### 5.1. Animals

All procedures involving horses were conducted according to animal welfare guidelines after approval by the Ethical Committee for the Use of Animals of the Federal University of Minas Gerais (CEUA/UFMG), under protocol number 159/2019, approved on 4 June 2018.

Eleven healthy crossbred horses, both male (n = 8) and female (n = 3), weighing 498.64 ± 72.62 kg and aged five to eight years old were used. Horses were kept in pasture and had access to 6 kg of alfalfa hay, 2 kg of a 12% protein commercial equine ration, plus 2 kg of hydrated/germinated oat grains per day. Mineral salt and freshwater were also provided ad libitum. Before engaging in immunization protocols, each horse was dewormed with drugs that were more appropriate to the species found in the parasitological exams (fenbendazole, ivermectin, albendazole, or moxidectin with praziquantel), and all of them were vaccinated against tetanus (using tetanus toxoid), leptospirosis, strangles, influenza, rabies, and encephalomyelitis. All horses had not hitherto partaken in any immunological protocol or experiment and were acquired from certified properties free of glanders and infectious equine anemia.

### 5.2. Venom and Immunization Protocol

Venom was obtained from spiders captured within Paraná and Santa Catarina states, Brazil, and kept under controlled conditions in the Centro de Produção e Pesquisa de Imunobiológicos (Production and Research Center of Immunobiological Products), State Department of Health, Piraquara, PR, Brazil. Specimens of *L. intermedia*, *L. gaucho*, and *L. laeta* were restrained from feeding for 30 days and then their venom was extracted using electrical stimulus of 12 V applied on the cephalothorax region. The venom pool obtained was lyophilized and kept on −20 °C, in the dark, until its use. Electrophoretic profile of L. intermedia venom was performed on SDS-polyacrylamide gel [38] is illustrated on Figure 3.

Horses followed an immunization schedule shown in Table 5, reaching an antivenom procurement level after 221 days. The immunogen was injected subcutaneously in four different points of neck and gluteal regions, alternating the injection sites to minimize the local reactions. Horses were clinically examined, underwent electrocardiographic recording (ECG), and had their blood sampled at three different moments: T0 (before day 1 of the cycle), T1 (day 115; 114 days after their first hyperimmunization and two days after reimmunization); and T2 (day 225; four days after the first bleeding for antivenom procurement). Sampling for biological quality control took place on days 38, 113, and 207 of the immunization cycle. This sampling was taken by blood withdrawn from the jugular vein using a vacuum tube containing clot activator and was immediately sent for analyses regarding its venom neutralization assay in mice and antibody titration. After an adequate antibody titration, the bleeding for antivenom procurement was then performed, in which 7 L of blood was sampled three times every other day, adding to 21 L of total volume. Bleeding for antivenom procurement took place on days 214, 216, and 221 of the cycle. The horses were monitored by clinical examinations and by hematocrit.

### 5.3. Clinical and Hematological Examinations

Horses were restrained using a halter and examined inside a stock. No sedation was needed. Horses underwent a thorough physical examination, encompassing inspection, measurement of parametric indexes, and clinical evaluation of biological systems [31].

Blood sampling was performed after the antisepsis of the external jugular vein region using 70% alcohol. Vacuum tubes containing ethylenediaminetetraacetic acid (EDTA) or clot activator (BD Vacutainer, Becton Dickinson, Curitiba, PR, Brazil) were used to perform hematologic and biochemical analyses, respectively. Hematological analyses were performed using an automatic cell counter (pocH-100Iv-Diff, Sysmex, São Paulo, SP, Brazil) calibrated for horse parameters. Serum biochemistry was determined by automatized equipment (Cobas Mira Plus, Roche, Montclair, NJ, EUA). Blood parameters evaluated were as follows: red blood cell count (RBC); hemoglobin; packed cell volume (PCV); white blood cell count (WBC); red blood cell distribution width (RDW); lymphocytes and sum of other WBCs, such as neutrophils, monocytes, and basophils (OTH); total platelet count (PLT); mean platelet volume (MPV); platelet distribution width (PDW); platelet clump (P-LCR); urea; creatinine; alanine aminotransferase (ALT); aspartate transaminase (AST); alkaline phosphatase (ALP); gamma-glutamyl transferase (GGT); glucose; total proteins (TP); albumin; globulins; cholesterol; triglyceride; lactate; and lactate dehydrogenase (LDH).

### 5.4. Electrocardiographic Evaluation

Horses underwent an ECG evaluation using a portable 12-channel digital electrocardiograph (TEB ECG Vet, Tecnologia Eletrônica Brasileira, São Paulo, SP, Brazil). ECG recordings were acquired in a quiet environment and with horses in an orthostatic position. Electrodes were fixed using alligator metal clips embedded in alcohol and attached to the horses’ skin using Dubois configuration, in which electrodes 1 and 2 were placed next to the spine tuberosity of both left and right scapula, electrode 3 on the xiphoid process of the sternum and electrode 4 on the proximal cranial region of the left forelimb [39]. Recordings were made at 25 mm/s speed and sensitivity of 1 cm = 1 mV. Bipolar (DI, DII, DIII) and augmented unipolar (aVR, aVL, aVF) leads were recorded. The following parameters were evaluated: cardiac rhythm; cardiac frequency; P (ms); P (mV); PR, QRS, and QT intervals; R and T waves; and ST segment levels. The electric axis was determined by QRS polarity in the bipolar and unipolar augmented leads.

### 5.5. Statistical Analysis

Statistical analysis was carried out using the SAS (version 9.0) software program. The obtained data were statistically analyzed using a mixed linear model approach of SAS (PROC MIXED), using first-order autocorrelation covariate structure. Animals were considered as a random factor with repeated measurements over time. *p* values < 0.05 were considered to indicate significance.

## Figures and Tables

**Figure 1 toxins-14-00338-f001:**
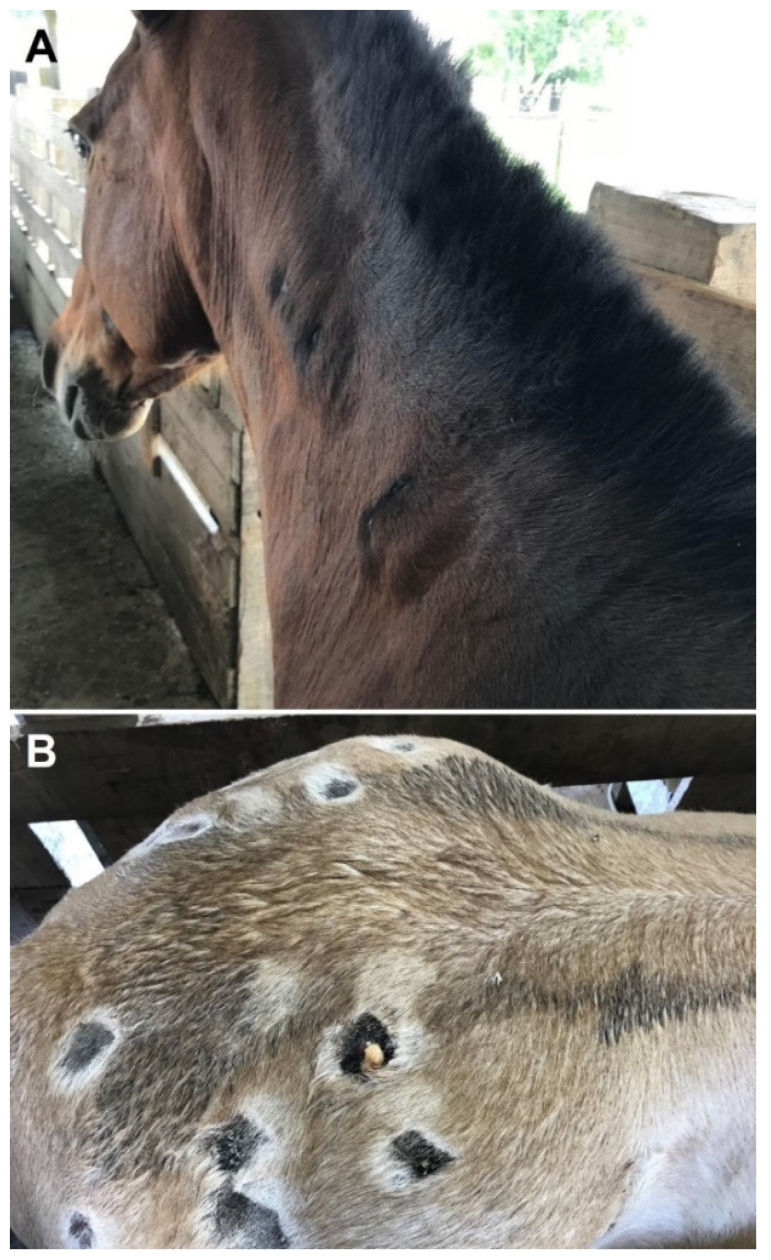
Abscesses in the infiltration sites of neck (**A**) and gluteal (**B**) regions of horses used for production of *Loxosceles* antivenom.

**Figure 2 toxins-14-00338-f002:**
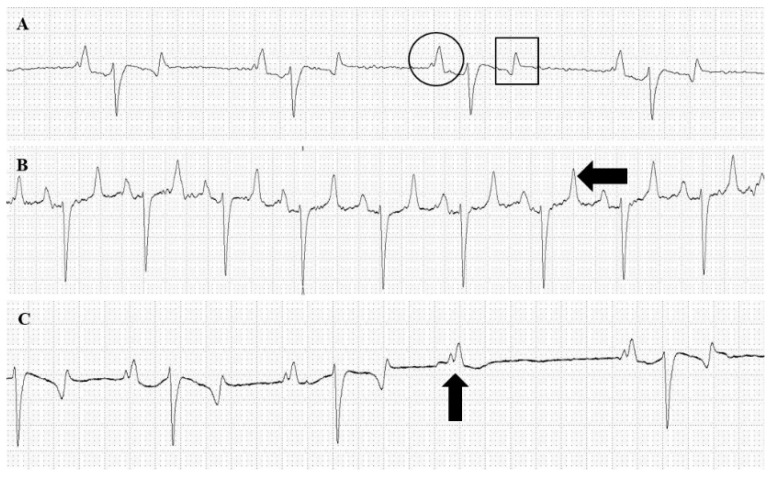
Electrocardiogram (ECG). (**A**) Normal sinus rhythm ECG from a horse before immunization. Common findings in ECG recordings were bifid P (circle) wave and biphasic T wave (square). (**B**) Sinus tachycardia from a horse after first bleeding. An increase in T wave was also observed (arrow). (**C**) Second-degree atrioventricular block from a horse after first bleeding. P wave (arrow) with no following QRS complex is observed.

**Figure 3 toxins-14-00338-f003:**
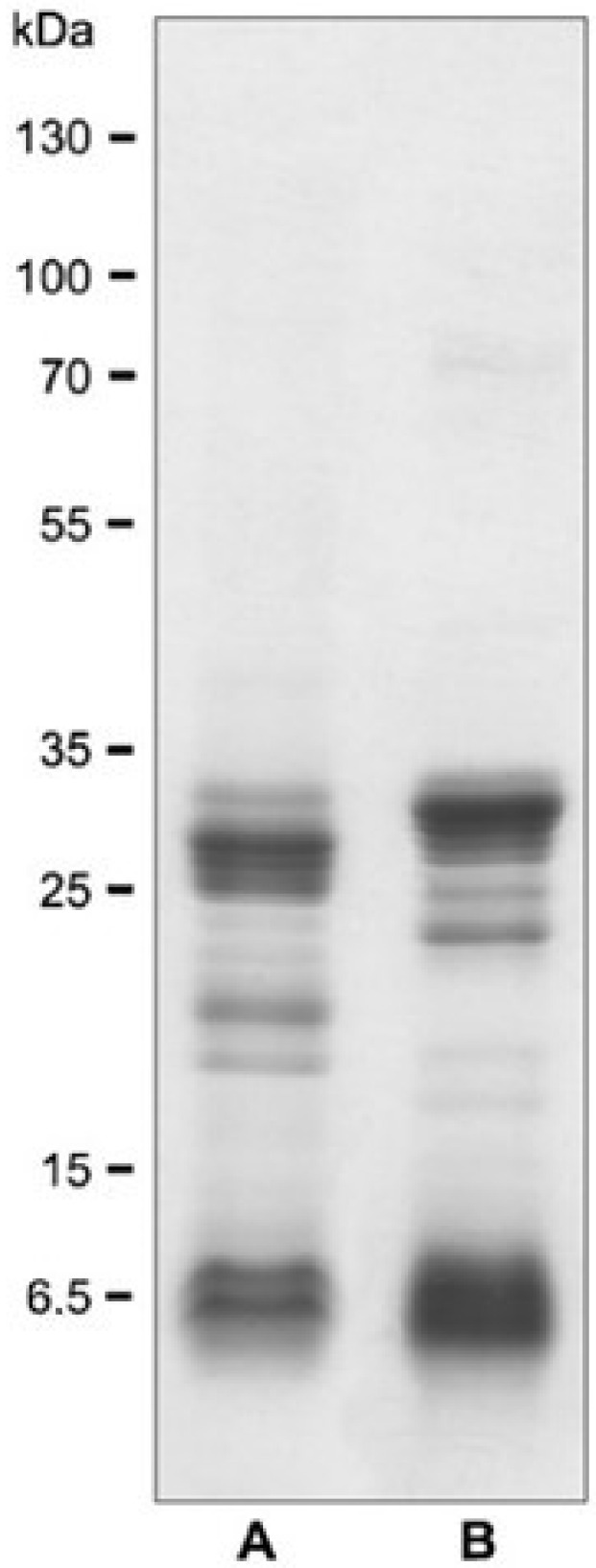
SDS-polyacrylamide gel (12.5%) electrophoretic separation of *L. intermedia* venom used in the study. (**A**) A non-reduced sample and (**B**) a reduced sample. Numbers on the left correspond to the positions of molecular weight markers (in kDa).

**Table 1 toxins-14-00338-t001:** Body weight, pulse and respiratory rates, and rectal temperature of horses that underwent immunization with *Loxosceles* venom examined before immunization (T0), after first venom immunization (T1), and after first bleeding (T2). Data are presented as mean ± standard deviation.

Clinical Parameter	T0	T1	T2
Body weight (kg)	509.0 ± 46.5	498.6 ± 76.2	507.4 ± 45.6
Pulse rate (ppm)	51.5 ± 11.2	49.4 ± 9.41	53.7 ± 12.6
Respiratory rate (mpm)	20.8 ± 5.88	22.3 ± 5.61	19.0 ± 16.5
Rectal temperature (°C)	38.0 ± 0.81	38.3 ± 0.22	37.8 ± 0.49

No significant difference observed (*p* > 0.05; regression analysis by mixed linear model).

**Table 2 toxins-14-00338-t002:** Body weight, pulse and respiratory rates, and rectal temperature of horses that underwent immunization with *Loxosceles* venom examined before immunization (T0), after first venom immunization (T1), and after first bleeding (T2). Data are presented as mean ± standard deviation.

Parameter	T0	T1	T2	Reference Values
RBC (cell × 10^6^/μL)	8.67 ± 1.30 ^a^	6.77 ± 1.00 ^b^	4.67 ± 1.68 ^c^	6.4–10.0
Hemoglobin (g/dL)	13.3 ± 2.30 ^a^	10.3 ± 1.78 ^b^	7.30 ± 3.14 ^c^	11.0–17.0
PCV (%)	39.8 ± 6.90 ^a^	31.4 ± 5.90 ^b^	21.91 ± 5.35 ^c^	32.0–47.0
RDW-SD (fL)	38.0 ± 2.25	39.1 ± 3.07	39.6 ± 3.01	-
RDW-CV (%)	20.9 ± 1.03 ^b^	21.7 ± 1.11 ^a^	21.8 ± 1.39 ^a^	21.0–25.0
WBC (cell × 10^3^/μL)	11.4 ± 2.76 ^b^	17.1 ± 3.93 ^a^	15.1 ± 4.10 ^a^	5.20–13.9
Lymphocytes (%)	33.5 ± 8.86 ^a^	18.7 ± 5.25 ^c^	28.8 ± 6.75 ^b^	-
OTHR (%)	66.5 ± 8.86 ^b^	81.3 ± 5.25 ^a^	61.1 ± 21.5 ^b^	-
Eosinophils (%)	0 ± 0	0 ± 0	0 ± 0	-
Lymphocytes (cell × 10^3^/μL)	3.68 ± 1.21 ^a^	3.13 ± 1.01 ^b^	4.19 ± 1.03 ^a^	1.5–7.7
OTHR (cell × 10^3^/μL)	7.75 ± 8.19	14.0 ± 3.58	10.9 ± 3.66	-
Eosinophils (cell × 10^3^/μL)	0 ± 0	0 ± 0	0 ± 0	-
PLT (cell × 10^3^/μL)	114.2 ± 53.6 ^b^	179.5 ± 50.6 ^a^	98.3 ± 78.3 ^b^	120.0–256.0
PDW (fL)	9.78 ± 0.95 ^a^	8.83 ± 0.44 ^b^	7.29 ± 1.00 ^c^	-
MPV (fL)	8.23 ± 0.40 ^a^	7.86 ± 0.26 ^b^	6.60 ± 0.35 ^c^	5.3–7.8
P-LCR (%)	6.70 ± 3.00 ^a^	4.57 ± 1.49 ^b^	0.60 ± 0.89 ^c^	-

RBC: Red blood cell count; PCV: packed cell volume; RDW: red blood cell distribution width; WBC: white blood cell count; OTHR: sum of other WBC (neutrophils, monocytes, and basophils); PLT: total platelet count; PDW: platelet distribution width; MPV: mean platelet volume; P-LCR: platelet clump. ^a,b,c^ Different letters in the same line show significant difference between evaluation periods (*p* < 0.05, regression analysis by mixed linear model).

**Table 3 toxins-14-00338-t003:** Blood biochemical panel of horses that underwent immunization with *Loxosceles* venom examined before immunization (T0), after first venom immunization (T1), and after first bleeding (T2). Data are presented as mean ± standard deviation.

Parameter	T0	T1	T2	Reference Values
Urea (mg/dL)	35.8 ± 4.22	36.3 ± 17.9	32.6 ± 3.83	21.4–51.5
Creatinine (mg/dL)	1.36 ± 0.35 ^a^	1.32 ± 0.31 ^a,b^	1.09 ± 0.17 ^b^	0.4–2.2
ALT (U/L)	4.32 ± 3.39	4.72 ± 2.89	5.29 ± 2.55	3.0–23.0
AST (U/L)	155.3 ± 33.7 ^a^	95.1 ± 36.0 ^b^	94.5 ± 28.9 ^b^	226–336
ALP (U/L)	159.3 ± 40.7	144.0 ± 57.1	164.4 ± 41.2	86.0–295.0
GGT (U/L)	10.4 ± 4.19 ^a,b^	17.4 ± 14.0 ^a^	5.94 ± 2.27 ^b^	6.0–32.0
Glucose (mg/dL)	101.3 ± 25.2 ^a^	72.8 ± 13.9 ^b^	103.5 ± 12.1 ^a^	62.0–134.0
TP (g/dL)	8.90 ± 1.46	9.24 ± 2.74	8.91 ± 0.81	6.0–8.0
Albumin (g/dL)	3.63 ± 0.73 ^a^	2.51 ± 0.36 ^b^	2.31 ± 0.24 ^b^	2.4–4.1
Globulins (g/dL)	5.28 ± 1.11	6.73 ± 2.54	6.60 ± 0.90	2.6–4.0
Cholesterol (mg/dL)	91.4 ± 18.6 ^a^	59.2 ± 7.23 ^b^	59.1 ± 6.36 ^b^	75.0–150.0
Triglycerides (mg/dL)	42.9 ± 9.56 ^b^	48.3 ± 8.74 ^b^	67.9 ± 12.6 ^a^	4.0–44.0
Lactate (mmol/L)	18.4 ± 5.33	37.7 ± 18.3	55.0 ± 97.3	10.0–16.0
LDH (U/L)	306.2 ± 127.3	253.5 ± 86.4	278.0 ± 84.8	162.0–412.0

ALT: Alanine aminotransferase; AST: aspartate transaminase; ALP: alkaline phosphatase; GGT: gamma-glutamyl transferase; TP: total proteins; LDH: lactate dehydrogenase. ^a,b^ Different letters in the same line show significant difference between evaluation periods (*p* < 0.05, regression analysis by mixed linear model).

**Table 4 toxins-14-00338-t004:** Electrocardiography parameters of horses that underwent immunization with *Loxosceles* venom examined before immunization (T0), after first venom immunization (T1), and after first bleeding (T2). Data are presented as mean ± standard deviation.

Parameter	T0	T1	T2	Reference Values [15]
Heart rate (bpm)	56.0 ± 12.6 ^a,b^	45.7 ± 4.29 ^b^	58.9 ± 12.4 ^a^	28–40
P (ms)	118.8 ± 21.6	126.5 ± 14.6	115.5 ± 19.4	<160
P (mV)	0.43 ± 0.08 ^a^	0.34 ± 0.09 ^b^	0.42 ± 0.07 ^a^	
PR (ms)	282.8 ± 68.4	293.7 ± 33.4	264.8 ± 57.3	<500
QRS (ms)	128.3 ± 23.0	139.0 ± 13.9	129.3 ± 13.1	<140
R (mV)	1.03 ± 0.50 ^b^	1.76 ± 0.58 ^a^	1.63 ± 0.56 ^a^	
QT (ms)	434.2 ± 62.3 ^b^	483.1 ± 33.9 ^a^	436.5 ± 46.8 ^b^	<600
T (mV)	0.46 ± 0.12 ^b^	0.79 ± 0.16 ^a^	0.76 ± 0.25 ^a^	

^a,b^ Different letters in the same line show significant difference between evaluation periods (*p* < 0.05, regression analysis by mixed linear model).

**Table 5 toxins-14-00338-t005:** Immunization protocol of naïve horses using *Loxosceles intermedia*, *L. laeta*, and *L. gaucho* venom.

Immunization Status	Day of the Cycle	Total Venom Amount	Venom Amount Per *Loxosceles* Species	Saline ^1^ Amount	Adjuvant
T0	0	Clinical examination, blood sampling, ECG recordings			
Hyperimmunization	1	300 µg	100 µg	1200 µL	1500 µL CFA ^2^
Hyperimmunization	11	450 µg	150 µg	1050 µL	1500 µL IFA ^3^
Hyperimmunization	22	750 µg	250 µg	2250 µL	No
Hyperimmunization	31	750 µg	250 µg	2250 µL	No
Sampling for biological quality control	38				
54-day rest					
Reimmunization	92	5 mg	1.67 mg	No	5 mL IFA
Reimmunization	106	5 mg	1.67 mg	5 mL	No
Reimmunization	113	5 mg	1.67 mg	5 mL	No
Sampling for biological quality control	113				
T1	115	Clinical examination, blood sampling, ECG recordings			
74-day rest					
Reimmunization	187	5mg	1.67 mg	No	5 mL IFA
Reimmunization	201	5 mg	1.67 mg	5 mL	No
Reimmunization	207	5 mg	1.67 mg	5 mL	No
Sampling for biological quality control	207				
Bleeding for antivenom procurement	214 216 221				
T2	225	Clinical examination, blood sampling, ECG recordings			
Antivenom procurement					

^1^ 0.85% NaCl. ^2^ CFA: complete Freund’s adjuvant. ^3^ IFA: incomplete Freund’s adjuvant.

## Data Availability

The datasets generated for this study are available on request from the corresponding author.

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
