# Peer review of "Clinical Effects of the Immunization Protocol Using Loxosceles Venom in Naïve Horses"

_toxins, 2022, doi:10.3390/toxins14050338_

Round 1

Reviewer 1 Report

The manuscript describes clinical effects observed in horses immunized with Loxosceles venom. The purpose is clear. The methods are sound. The results are interesting and important in the field. I found only minor concerns described below. Therefore, I recommend minor revision of the manuscript for the acceptance.

  1. “p” for “p value” is recommended to be italicized.
  2. Space should be inserted before and after inequality symbols for example p < 0.05, not p<0.05.
  3. Table 2, the expressions of statistical significance (a, b, c) are confusing in some cases. For example, T0 and T2 are labeled as “b” while T1 is labeled as “a” in OTHR In this case the statistical significance was observed between T0 vs T1 and T1 vs T2 individually? Otherwise T1 versus the pooled data of T0 and T2?? I suggest showing which pair of groups showed significance more clearly.

Author Response

The manuscript describes clinical effects observed in horses immunized with Loxosceles venom. The purpose is clear. The methods are sound. The results are interesting and important in the field. I found only minor concerns described below. Therefore, I recommend minor revision of the manuscript for the acceptance.

RESPONSE: We appreciate your comments.

1. “p” for “p value” is recommended to be italicized.

RESPONSE: Corrected in manuscript.

2. Space should be inserted before and after inequality symbols for example p < 0.05, not p<0.05.

RESPONSE: Corrected in manuscript.

3. Table 2, the expressions of statistical significance (a, b, c) are confusing in some cases. For example, T0 and T2 are labeled as “b” while T1 is labeled as “a” in OTHR In this case the statistical significance was observed between T0 vs T1 and T1 vs T2 individually? Otherwise T1 versus the pooled data of T0 and T2?? I suggest showing which pair of groups showed significance more clearly.

RESPONSE: We used the standard indication with a being the highest value, and c being the lowest. The indication of statistical differences in tables was reformulated.

Reviewer 2 Report

I think this is a very nicely done paper. It is educational about the innoculation process and its challenges and It should be accepted. 

Author Response

I think this is a very nicely done paper. It is educational about the innoculation process and its challenges and It should be accepted.

RESPONSE: We appreciate your comments.

Reviewer 3 Report

The authors present a prospective investigation of the clinical effects of Loxosceles venom in venom naïve horses. The 11 horses had measurements of vital signs, EKG, metabolic parameters and hematological parameters.

It appeared that the immunization protocol resulted is significant morbidity as documented by abscess formation in the sites of immunization with venom and thrombophlebitis in neck vessels. The horses had about half of the RBCs present after the protocol, accompanied by a persistent increase in WBC and decrease in platelets. The pattern is consistent with either bone marrow suppression or ongoing intravascular coagulation and infection. The metabolic parameters were also worrisome, with a decrease in albumin synthesis and near tripling of lactate consistent with hepatic dysfunction despite the lack of increased liver enzyme release. The increase in lactate could also be secondary to inadequate oxygen delivery to tissues with the dramatic loss of RBCs, which would also compromise the capacity of the liver to metabolize the lactate.

The myocardial signs could be secondary to primary venom-mediated toxicity or ischemia secondary to severe anemia.

I have a number of comments.

The authors should indicate in the first few lines of results what the time period was between the time points (e.g., treatment duration in days + rest days).

Table 2-3. It is unclear what a, b and c indicate. Please be more explicit/descriptive.

Line 83. “first cbleeding” is a typo.

Figure 3. This figure is presented, but no attempt is made to identify what the bands represent. Also, the methodology used to generate figure 3 are not included.

Statistics. The methods used seem inappropriate. The authors should have used repeated measures one-way ANOVA with a post hoc test such as Tukey or SNK.

In sum, this is an interesting paper about an important issue. I would like a stronger conclusion to reflect that.

Author Response

The authors present a prospective investigation of the clinical effects of Loxosceles venom in venom naïve horses. The 11 horses had measurements of vital signs, EKG, metabolic parameters and hematological parameters.

It appeared that the immunization protocol resulted is significant morbidity as documented by abscess formation in the sites of immunization with venom and thrombophlebitis in neck vessels. The horses had about half of the RBCs present after the protocol, accompanied by a persistent increase in WBC and decrease in platelets. The pattern is consistent with either bone marrow suppression or ongoing intravascular coagulation and infection. The metabolic parameters were also worrisome, with a decrease in albumin synthesis and near tripling of lactate consistent with hepatic dysfunction despite the lack of increased liver enzyme release. The increase in lactate could also be secondary to inadequate oxygen delivery to tissues with the dramatic loss of RBCs, which would also compromise the capacity of the liver to metabolize the lactate.

The myocardial signs could be secondary to primary venom-mediated toxicity or ischemia secondary to severe anemia.

RESPONSE: We appreciate your comments.

I have a number of comments.

The authors should indicate in the first few lines of results what the time period was between the time points (e.g., treatment duration in days + rest days).

RESPONSE: This period is detailed in Table 5.

Table 2-3. It is unclear what a, b and c indicate. Please be more explicit/descriptive.

RESPONSE: The indication of statistical differences in tables was reformulated.

Line 83. “first cbleeding” is a typo.

RESPONSE: Corrected in manuscript.

Figure 3. This figure is presented, but no attempt is made to identify what the bands represent. Also, the methodology used to generate figure 3 are not included.

RESPONSE: More information about the electrophoresis of the venom was included in manuscript. Unfortunately, it is not possible to identify the bands because this venom is a very complex solution that isn't fully understood.

Statistics. The methods used seem inappropriate. The authors should have used repeated measures one-way ANOVA with a post hoc test such as Tukey or SNK.

RESPONSE: In SAS, PROC MIXED is a powerful procedure that is used for analyzing longitudinal data, including repeated measures analysis of variance. You can actually perform repeated measures ANOVA with PROC MIXED.

In sum, this is an interesting paper about an important issue. I would like a stronger conclusion to reflect that.

RESPONSE: We have expanded the conclusions section.

Round 2

Reviewer 3 Report

No further comments.